# Large Improvements in Health-Related Quality of Life and Physical Fitness during Multidisciplinary Inpatient Rehabilitation for Pediatric Cancer Survivors

**DOI:** 10.3390/cancers14194855

**Published:** 2022-10-04

**Authors:** David Riedl, Thomas Licht, Alain Nickels, Maria Rothmund, Gerhard Rumpold, Bernhard Holzner, Vincent Grote, Michael J. Fischer, Gustav Fischmeister

**Affiliations:** 1Ludwig Boltzmann Institute for Rehabilitation Research, 1100 Vienna, Austria; 2University Hospital of Psychiatry II, Department of Psychiatry, Psychotherapy, Psychosomatics and Medical Psychology, Medical University of Innsbruck, 6020 Innsbruck, Austria; 3Pediatric Rehabilitation Center Leuwaldhof, 5621 St. Veit im Pongau, Austria; 4Oncological Rehabilitation Center, 5621 St. Veit im Pongau, Austria; 5Institute of Psychology, University of Innsbruck, 6020 Innsbruck, Austria; 6University Hospital of Psychiatry I, Department of Psychiatry, Psychotherapy, Psychosomatics and Medical Psychology, Medical University of Innsbruck, 6020 Innsbruck, Austria; 7VAMED Rehabilitation Center Kitzbuehel, 6370 Kitzbuehel, Austria

**Keywords:** pediatric cancer, HRQOL, pediatric rehabilitation, observational study, performance score

## Abstract

**Simple Summary:**

After cancer treatment, children often suffer from various health problems. Rehabilitation can improve their quality of life. We present data for 236 children and 478 parents who filled out questionnaires before and after inpatient rehabilitation. In our sample, both children and parents reported that the quality of life of the children had substantially improved during rehabilitation, especially their physical and psychosocial quality of life. Children also improved in physical exercises and their functional status. Parents and children showed better agreement regarding the child’s health status after rehabilitation than before. A specific method of statistical analysis (performance score) helped to reduce disagreement between parents and children. We therefore conclude that inpatient rehabilitation is effective for children after cancer treatment and that the performance score should be used in research on rehabilitation for children after cancer treatment to have the most accurate results.

**Abstract:**

Rehabilitation is a key element in improving health-related quality of life (HRQOL) for pediatric cancer survivors. The aim of this study was to present data from a multidisciplinary inpatient rehabilitation treatment. Children took part in a four-week multidisciplinary family-oriented inpatient rehabilitation. A total of 236 children (>5–21 years) and 478 parents routinely completed electronic patient-reported outcomes (ePROs), performance-based assessments, and clinician-rated assessments before (T1) and at the end (T2) of rehabilitation. HRQOL was assessed with the PedsQL generic core and PedsQL cancer module. Data were analyzed using repeated measures analysis of variance (ANOVA). Statistically significant improvements with medium to large effect sizes were observed for most HRQOL scales (η^2^ = 0.09–0.31), as well as performance-based and clinician-rated assessments for physical activity and functional status (η^2^ > 0.28). Agreement between children’s PROs and parents’ proxy ratings was lower before (r_ICC_ = 0.72) than after (r_ICC_ = 0.86) rehabilitation. While the concordance between children and parents’ assessment of changes during rehab was low to moderate (r = 0.19–0.59), the use of the performance score led to substantially increased scores (r = 0.29–0.68). The results of this naturalistic observational study thus highlight the benefits of multidisciplinary pediatric inpatient rehabilitation for childhood cancer survivors. The use of the performance score is recommended in this field.

## 1. Introduction

Over the past few decades, the worldwide incidence of childhood cancer has increased by 13% [1]. Each year, 9.1–17.8 cases (per 100,000 children) of childhood cancer are newly diagnosed in Europe [2]. Compared to adult oncology, children and adolescents represent a unique patient population with distinct features in physiological and cognitive development and are affected by different types of cancer. Three major pediatric cancer diagnoses account for approximately two-thirds of all childhood cancers, namely, leukemias (~30%), central nervous system (CNS) tumors (24–26%), and lymphomas (11–14%) [3,4]. While cancer remains the leading cause of disease-related death among children, the overall mortality has decreased by more than 50% over the past few decades, and overall survival rates have reached >80% [5,6].

Depending on cancer type, location, and stage, treatment may include surgery, chemotherapy, radiotherapy, stem cell transplantation, or immunotherapy. Although treatment regimens have been optimized, there is a significant number of acute and late adverse events, including physical impairment (such as fatigue, cardiotoxicity, and neurocognitive deficits) and decreased psycho-social functioning [7]. Long-term survivors of childhood cancer are at significant risk for treatment-related late effects for life [8,9].

Cancer rehabilitation should be considered a key tool to reach and maintain improved physical and psychosocial functioning [10]. Cancer rehabilitation usually consists of multidisciplinary in-patient or outpatient treatment, such as physiotherapy, psychotherapy, occupational therapy, nutrition counseling, and lifestyle interventions [11,12]. At its core, rehabilitation aims to improve bio-psycho-social aspects of the individual’s health status and health-related quality of life (HRQOL) [13]. Due to the subjective nature of many of those aspects, both the Food and Drug Administration (FDA) and the European Medicines Agency (EMA) have recommended the use of patient-reported outcome measures (PROMs) in pediatric oncology [14,15]. It is also recommended to complement children’s self-report with caregiver proxy measures [16,17]. Proxy reports are essential when children are neurocognitively impaired, too young, or too ill to respond for themselves [18,19], which is frequently the case in pediatric oncology.

However, the use of proxy reports is somehow problematic, as research has shown that parents consistently overestimated symptoms and underestimated functioning compared to the children’s self-report [20,21,22]. Factors that may influence parent proxy ratings include the child’s age and sex, level of parental education, cultural background, the parents’ own health, QOL, and psychological distress, as well as the parent’s sex [16,23,24,25,26,27]. So far, however, to our best knowledge, no study has investigated the association of patients’ self-reports and the respective parental proxy ratings in oncological rehabilitation for children.

While several studies from oncological rehabilitation in adult patients have shown the beneficial effects of multidisciplinary inpatient cancer rehabilitation [28,29,30,31], there is only limited evidence for rehabilitation in pediatric cancer survivors. Thus, in this study, we (1) investigated the effect of pediatric oncological rehabilitation on patients’ HRQOL, (2) investigated patient–proxy discrepancies in HRQOL ratings, and (3) compared different approaches to minimize discrepancies between patient PRO and parent proxy ratings. To this end, we conducted a longitudinal, single-center, observational study using self- and observer-reported HRQOL as well as functional health aspects in a sample of pediatric cancer survivors. We hypothesized that (a) the pediatric patients would show significant improvements in several aspects of HRQOL and functional health, and (b) children and parents would differ in their assessment of HRQOL. 

## 2. Materials and Methods

### 2.1. Patient Sample and Recruitment

Data were collected as part of the clinical procedures at the pediatric oncological rehabilitation center “Leuwaldhof” (Sankt Veit im Pongau, Austria). Pediatric cancer survivors took part in a multidisciplinary inpatient rehabilitation treatment. Treatment was also offered to accompanying family members, e.g., parents and siblings. In Austria, the costs for the rehabilitation treatment are covered by social insurance institutions. The typical rehabilitation treatment for pediatric cancer survivors lasted 28 days with 2.5 daily hours of different therapeutic interventions. 

Patients and accompanying parents completed the baseline assessment (T1) before being admitted to the rehabilitation clinic, and a second assessment (T2) at the end of their stay. Patients and/or legal guardians signed a written informed consent prior to data collection. Children between 5 and 21 years old completed the self-report version of the questionnaires, and parents independently completed a proxy version of the questionnaires. Data were collected electronically using a multifunctional web-based application called the Life App, which is based on Computer-Based Health Evaluation Software (CHES) [32]. Collected data were used to evaluate the rehabilitation outcomes, and the results were discussed with the patients at the end of the rehabilitation treatment. 

### 2.2. Rehabilitation Treatment

All patients received a set of multidisciplinary therapies including medical and nursing treatment, nutritional counseling, physiotherapy, and psychological therapies. Depending on their specific needs, patients also took part in speech therapy, massage, animal assisted therapy, social counseling, or pedagogical interventions. Patients received a median number of 85 therapy sessions (IQR = 79–95). The frequency of therapeutic units is displayed in Table 1. 

### 2.3. Patient-Reported Outcomes (PROs)

The assessment of health-related quality of life (HRQOL) was conducted using the Pediatric Quality of Life Inventory (PedsQL) 4.0 Generic Core Scales [33] and the PedsQL 3.0 Cancer Module [34]. The PedsQL 4.0 Generic Core Scales consists of 23 items, which assess five functioning subscales (physical, emotional, social, school, psychosocial), and a total score. The PedsQL 3.0 Cancer Module consists of 27 items to assess eight symptom subscales (pain and hurt, nausea, procedural anxiety, treatment anxiety, worry, cognitive problems, perceived physical appearance, and communication) and a total score [34]. Both questionnaires are available in age-specific versions and offer a patient self-report and an observer/proxy version. Higher values on both measures represent better outcomes (i.e., higher functioning and lower symptom load). The minimal clinically important difference (MCID) for the total score of the PedsQL 4.0 Generic Core Scales is 4.4 points for the child self-report version and 4.5 points for the parent proxy version [33].

As required by the Austrian Health Insurance and Pension Fund, individual rehabilitation goals are discussed with patients at the beginning of the rehabilitation treatment [35]. At the end of the treatment, rehabilitation outcome is evaluated by the health-care professionals and patients based on the individual treatment goals. Each goal is either rated as “successful”, “partially successful”, or “not successful”.

### 2.4. Observer-Reported and Performance Outcomes

The Karnofsky Performance Scale Index is a tool for observer-based assessments of the functional impairment of patients. Impairment of the patients’ activity, autonomy, and self-care abilities was rated on a 0–100% spectrum, with higher scores indicating better levels of functioning. In our study, the score was summarized as follows: high level of functioning (80–100%), medium level of functioning (50–80%), and low level of functioning (0–50%). To evaluate the physical fitness, a six-minute walk test (6-MWT) was applied at the beginning and end of the rehabilitation stay for children who were old enough and physically developed enough to perform the test. Body mass index (BMI) was extracted from the clinic’s information system.

### 2.5. Statistical Analysis

Differences in mean PedsQL scores between child self-assessment and parent proxy ratings were tested for statistical significance using independent sample *t*-tests. Multivariate analyses of variance (MANOVA) applying Pillai’s Trace were applied to compare baseline scores of patients with complete datasets, as well as to investigate differences in PedsQL baseline scores across cancer types. Changes in HRQOL (PedsQL), walking distance (6-MWT), and functional status (Karnofsky Index) between the two time points were analyzed using repeated measures ANOVAs. For the PedsQL, analyses were conducted separately for the patient self-reports and parent proxy reports. The frequency of individual treatments is presented with median and interquartile ranges (IQR). Partial eta-squared (*η^2^*), Cohens’ d, and Phi (ϕ) were calculated as measures of effect sizes. Effect size values of d = 0.3, *η^2^* = 0.01, and ϕ = 0.1 were considered small; d = 0.5, *η^2^* = 0.06, and ϕ = 0.3 were medium; and d = 0.8, *η^2^* = 0.14, and ϕ = 0.5 were large [36]. 

Concordance and differences between patient-reported outcomes (i.e., children’s self-assessment) and the parent’s proxy ratings were evaluated by calculation of interclass correlation coefficients (r_ICC_; two-way mixed, absolute agreement) for all PedsQL subscales separately. To test the overall agreement between the children’s and parent’s ratings, we applied z-transformation to all PedsQL subscales at T1 and T2 separately for the child and parent ratings and calculated a mean score across both measurement points for children and parents. The correlation of these z-transformed mean scores for children and parents was then also analyzed with r_ICC_. Values of r_ICC_ less than 0.5, between 0.5 and 0.75, between 0.75 and 0.9, and greater than 0.90 were interpreted as poor, moderate, good, and excellent reliability, respectively [37]. It has been previously reported that due to baseline differences the calculation of change scores may lead to ceiling effects and thus may distort the interpretation of therapy effectiveness across groups with differences in baseline values. If, for example, a patient already reports good HRQOL scores before rehabilitation, no substantial increase (change) is expected during rehabilitation. In this case, although the patient’s overall performance may be good, only a slight or no increase can be measured by simply calculating the mean difference. A simple solution might be the use of the “performance score (T2D)”, based on the formula *T2* + (*T2* − *T1*), which reflects the individual performance and considers the functional status at the beginning of rehabilitation (changes from *T1* to *T2*; Δ) without problems of mathematical coupling or regression effects, as seen in ANCOVA [31,38]. To test this approach, associations between the assessment of children and their parents for the mean differences between T1 and T2 (delta, Δ) as well as the performance score (T2D) were compared with Pearson correlation coefficients. Values of r < 0.3 were interpreted as negligible, while r = 0.3–0.5 indicated low, 0.5–0.7 moderate, 0.7–0.9 high, and >0.9 an excellent correlation [39,40,41]. To compare mean r-values of delta and T2D, r-scores were transformed to normally distributed variables z’ by Fisher z-transformation, and mean z-scores were transformed back to mean r-values. *p* values of <0.05 were considered statistically significant. All calculations were conducted with SPSS (v21).

## 3. Results

In total, *n* = 236 cancer survivors were included in the analyses. The mean age of the patients was 11 (SD: 4.3) years and 58.9% were male. The most frequent diagnoses were leukemias (35.2%), lymphomas (19.5%), and brain tumors (13.6%), and almost all patients (98.4%) had a medium to high functional status (Karnofsky Performance Score). The proxy sample consisted of *n* = 478 caregivers with a mean age of 42.1 (SD: 7.3) years; the majority of the sample was female (*n* = 306, 66.4%). The mean interval between admission to and discharge from the rehabilitation center was 28.0 (SD = 3.6) days. More details can be found in Table 2.

Compete datasets were available for *n* = 236 children at T1 and *n* = 182 children at T2 (77.1%). For parents, complete datasets were available for *n* = 478 participants at both assessment points. A total of *n* = 149 datasets with children’s PROs and parents proxy ratings (63.4%) could be matched.

### 3.1. HRQOL Reports Prior to Rehabilitation

We analyzed data from T1 to evaluate the patients’ health status prior to the rehabilitation treatment. Based on the children’s self-reports, the worst scores were found for procedural anxiety and school functioning, while for treatment anxiety, the best overall scores were reported. Parents, on the other hand, reported the lowest scores for their children for procedural anxiety, communication, and emotional functioning. Overall, parents reported significantly worse HRQOL on all scales except school functioning (diff = 2.0 points; t = 13.89, *p* = 0.17), treatment anxiety (diff = 1.4 points; t = 0.70, *p* = 0.48), and perceived physical appearance (diff = 2.6 points; t = 13.57, *p* = 0.18). The largest differences between the children’s self-assessment and parent proxy ratings were found for communication (diff = 20.6 points; t = 96.1, *p* < 0.001; d = 0.77), procedural anxiety (diff = 12.3 points; t = 43.68, *p* < 0.001, d = 0.34), and emotional functioning (diff = 11.6; t = 74.3, *p* < 0.001, d = 0.6), with medium to large effect sizes.

We also analyzed differences in the PedsQL subscales across the different cancer entities at baseline. Based on the children’s self-assessment, no significant differences were observed for any subscales (F (6, 225) = 1.09, *p* = 0.28). However, parents reported significant differences across the baseline PedsQL scores for different cancer entities regarding pain (F (6, 245) = 3.53, *p* = 0.002). Post-hoc tests showed that parents reported significantly worse pain scores for children with sarcoma (67.3 points; *p* = 0.004–0.043), leukemia (70.0 points; *p* < 0.001–0.011), brain tumors (72.7 points; *p* = 0.005), and malignant lymphomas (72.9 points; *p* = 0.010) compared to children with neoplasms of blood and immune system (83.9 points) and children with other malignancies (89.9 points). Accordingly, children with sarcomas (mean: 82.5%) had significantly lower overall Karnofsky Performance Scores compared to patients with malignant lymphomas (mean: 90.0%; *p* = 0.012) and leukemias (*p* = 0.003; 90.2%). No statistically significant differences were observed between the other cancer entities. 

### 3.2. Improvement in HRQOL during Rehabilitation Treatment

#### 3.2.1. Child Self-Report

Based on the children’s self-report, statistically significant improvements were observed for all functioning scales and most symptom scales at the end of the rehabilitation, except for treatment anxiety, worries, and perceived physical appearance. The largest effect sizes for improvements were found for physical functioning (diff = 7.5 points), psychosocial functioning (diff = 5.0 points), and school functioning (diff = 5.5 points). For details, see Table 3.

#### 3.2.2. Proxy Ratings

Based on the proxy ratings, statistically significant improvements were observed for all variables, except communication. Generally, proxy ratings showed larger improvements than the children’s self-ratings. The largest effect sizes for improvements were found for emotional functioning (diff = 9.6 points), psychosocial functioning (diff = 7.5 points), and physical functioning (diff = 7.3 points). For details, see Table 4.

### 3.3. Improvement in Health Status and Goal Achievement during the Rehabilitation

A mean number of 3.7 goals (SD: 1.0) per patient were agreed on at the beginning of the rehabilitation treatment, and the number of individual goals ranged from 1 to 8 goals. The vast majority of patients (*n* = 185; 78.4%) had met all their individual goals during rehabilitation, and no patient reported that they did not meet any goals at all. Patients also showed a significant improvement in the mean walking distance at the end of rehabilitation compared to admission, with a large effect size (515.6 vs. 566.4 m; F (1, 160) = 70.30, *p* < 0.001, η^2^ = 0.31), as well as an improved observer-rated functional status (Karnofsky: 88.4% vs. 92.9%; F (1, 216) = 82.60, *p* < 0.001, η^2^ = 0.28), also with a large effect size. No statistically significant change was observed for the mean BMI (19.9 vs. 19.8; F (1, 115) = 1.14, *p* = 0.29, η^2^ = 0.01).

### 3.4. Patient–Proxy Discrepancies

In our sample, n = 149 dyads of complete PRO and proxy assessments were available at baseline (T1). Of these, 95/149 included proxy ratings from both parents and 54/149 from only one parent. The mothers’ assessments were available in 142/149 dyads, while 102/149 father–child dyads were available. At the end of rehabilitation (T2), *n* = 103 dyads were available. Of these 59/103 included proxy ratings from both parents and 42/103 from only one parent. The mothers’ assessments were available in 96/103 dyads, while 66/103 father–child dyads were available.

Overall agreement between children and parent ratings using z-transformed mean scores across both measurement points was good (r_ICC_ = 0.81; 95% CI: 0.73–0.87). However, when analyzing each time point separately, agreement for T1 (r_ICC_ = 0.72; 95% CI: 0.60–0.81) was lower than for T2 (r_ICC_ = 0.86; 95% CI: 0.79–0.91).

At the beginning of the rehabilitation treatment, child–mother dyads showed moderate correlations (r_ICC_ = 0.53–0.69) for functioning scales, with the highest agreement for physical functioning. For child–father dyads, similar results were found (r_ICC_ = 0.57–0.65). Regarding the assessment of symptoms, poor to moderate correlations were found between child–mother dyads (r_ICC_ = 0.49–0.84). They were, however, consistently higher than the child–father dyads (r_ICC_ = 0.40–0.78). In both dyads, the highest agreement was found for rating of procedural anxiety, while the communication (with health-care professionals) scale revealed no significant association in either dyad. Agreement between both parents was generally good (r_ICC_ = 0.76–0.86), except concerning cognitive problems (r_ICC_ = 0.61) and communication (r_ICC_ = 0.65). For details, see Table 5.

At the end of rehabilitation, the general agreement for functioning scales increased substantially in both child–mother (r_ICC_ = 0.77–0.88) and child–father dyads (r_ICC_ = 0.76–0.85), yielding good overall agreement between both parents and the child, as well as between both parents. 

As for the symptom ratings, overall agreement between both parents and their child was also substantially better than before rehabilitation, with good to excellent agreement, except for cognitive problems (r_ICC_ = 0.40–0.64) and communication (r_ICC_ = −0.36–0.01). When separately analyzing both parents’ agreement with their child, fathers showed higher agreement regarding pain, nausea, procedural anxiety, and treatment anxiety, while mothers showed higher agreement regarding worries and perceived physical appearances. For details, see Table 6.

### 3.5. Accordance of Mean HRQOL Change Ratings by Children and Parents

To further evaluate the agreement between children and parents regarding mean change in HRQOL during rehabilitation, we evaluated the percentage of children responding with clinically important improvements (i.e., individual score changes above MCID) on the PedsQL Generic total score, based on the children’s self-report and parental proxy ratings for each cancer type separately. In general, while children reported lower rates of clinical improvement than their parents (53.3% vs. 58.4%), this difference was not statistically significant (χ^2^ = 1.36, *p* = 0.24; ϕ = 0.05). Children with soft tissue neoplasms (64.3%) and bone cancers (56.5%) reported the highest rates of clinically relevant improvements, while the lowest rates were found for patients with brain cancer (48.0%). Based on the parent proxy ratings, the highest rates were found for patients with malignant lymphomas (75.0%) and bone cancer (73.3%). However, the percentage of clinically relevant improvement did not significantly differ between tumor groups, either for children’s self-report (χ^2^ = 2.02, *p* = 0.92; ϕ = 0.11) nor for parents’ proxy ratings (χ^2^ = 7.31, *p* = 0.29; ϕ = 0.16). For details see Figure 1.

Secondly, we tested two further approaches to evaluate mean changes during rehabilitation: calculation of delta (i.e., mean difference), and performance score (adjusted delta). When comparing mean delta values, low to moderate agreement between children and parents was observed (r = 0.20–0.59), with generally better agreement for functioning scales (r = 0.59) than for symptom scales (r = 0.45). While the highest agreements were observed for psychosocial functioning and physical functioning, no significant association was found for nausea, worries, communication, or the overall symptom rating. For details, see Table 7.

A comparison of the children’s and parents’ performance scores yielded substantially improved agreement for all assessed subscales, with at least moderate correlations (r = 0.29–0.68). For functioning scales, the largest improvements were found for emotional functioning, social functioning, and school functioning. However, improvements in accordance were substantially larger regarding the symptom subscales, with the largest improvement for procedural anxiety, treatment anxiety, and nausea. For all subscales, except for communication, a significant association between the children’s self-assessment and parental proxy ratings could be observed. In summary, the mean correlation between children and parents was significantly higher for the T2D performance score than for the delta (r = 0.50 vs. r = 0.34; z = −1.668; *p* = 0.048). For details, see Table 7.

## 4. Discussion

The aim of this study was to analyze changes in HRQOL and observable indicators of physical functioning in children after cancer treatment during the course of a four-week multidisciplinary inpatient rehabilitation treatment. To our knowledge, this is the largest study on inpatient rehabilitation for pediatric cancer survivors analyzing both self-reported and observer/proxy measures of health. We observed large and clinically meaningful improvements across the most assessed functional aspects of HRQOL and observable indicators of physical functioning. The use of the performance score led to a substantially better concordance between the children’s self-assessment and parental proxy ratings.

Since studies have shown that children with cancer and their parents commonly disagree on less observable aspects of HRQOL [16,23,42], we analyzed both the children’s self-report as well as their parents’ proxy reports to gain a full picture of the children’s health. At baseline, children reported most problems with anxiety related to medical procedures (i.e., procedural anxiety) and school-related factors, while parents additionally highlighted emotional problems and issues with communicating about their disease. Previous studies show that the vast majority of childhood cancer survivors (>85%) have to face at least one form of late effect, such as problems with cognition, body image, fatigue, chronic pain, or depression, that interfered with their ability to participate in major life activities [43]. These issues are associated with several long-term consequences. In a large-scale study that investigated the needs of >12,000 survivors of childhood cancer, school problems were frequently reported. Almost a fourth of the sample had to use special education services, and the likelihood of finishing school or college was significantly reduced for children with various cancer types when compared to their siblings [44]. Poor mental health among childhood cancer survivors was associated with lower income or unemployment and a reduced likelihood of marriage [45]. Problems with communication have been identified as a common problem among children with cancer, and merit specific interventions targeting language skills [46], since they may further increase the risk of unemployment [47]. This may explain the high uptake of speech therapy (>87%) in our sample. In regard to type of malignancy, we found no statistically significant differences for any of the children’s self-assessed PedsQL scales at baseline. However, when comparing parent rated PedsQL scales as well as the Karnofsky Performance score, patients with sarcomas showed significantly worse pain and performance scores than patients with other malignancies. This is in line with previous research, which has found parent-reported cancer specific HRQoL impairment after treatment in children with cancer [48].

Children and their parents showed substantial discrepancies in their assessment of the child’s functional status, especially regarding communication, procedural anxiety, and emotional functioning. Parents generally reported lower HRQOL scores at baseline than the children themselves. Comparisons of clinician-rated performance status, parental proxy ratings, and children’s self-reports indicate that differences between reporters are most likely a matter of perspective, rather than insufficient internal consistencies of the measures [49]. In contrast to a recent study on children during active chemotherapy [27], we found rather good agreement between mothers’ and fathers’ proxy ratings of their child and the children’s self-assessment. However, the observed agreement was substantially better at the end of rehabilitation treatment than before the treatment. One possible explanation is that the parents proxy ratings are connected to the children’s health status, i.e., that parents’ observations of their child’s HRQOL are more accurate if the child is healthier. This would also explain why accordance between parents is better in cancer survivors than in children during active chemotherapy [27,50]. This aspect may be analyzed in more detail in future studies. Discrepancies between parental ratings and the children’s self-reports may be influenced by several factors, including the parents’ own psychological distress [51,52]. It thus may be considered a strength of family-oriented inpatient rehabilitation that (psychological) treatment can be offered not only to the child, but to the whole family system [53].

Discordance between children’s PROs and parental proxy ratings are problematic in pediatric oncology since both approaches are frequently used in combination. However, so far, no guidelines offer recommendations to clinicians or researchers concerning how to handle discrepant results. In our study, for example, children reported no significant mean improvement in treatment anxiety, while parents observed a statistically highly significant difference with a moderate effect size. If this subdomain had been a priori defined as a primary or secondary outcome, it would not be clear how to interpret the results. Therefore, we also tested the so-called “performance score”, which was recently introduced as a more robust method to compare health assessments, taking into account different individual baseline results [38]. By correcting the delta for the T2 scores, the performance score is supposed to improve interpretability of change scores across different assessment types or groups with differing baseline scores. However, it has to be highlighted that while the T2D may help to identify real differences (positive and negative or neutral) and to reduce bias associated with delta scores, this does not automatically result in more ‘positive’ results. In our study, we found a substantially increased accordance between children’s self-assessment and parental proxy ratings when applying the performance scores. Additionally, while based on the mean change (delta) solely, we found no significant correlation between the child’s self-assessment of change and parental proxy ratings for nausea, worries, and communication based on the use of the performance score. Only the communication subscale showed no significant association. Based on these preliminary findings, we recommend further research on the use of the performance score in pediatric oncology and rehabilitation research.

In our sample, the largest improvements during treatment were observed for physical functioning and psychosocial functioning from both children and parents. The effect sizes were comparable to research from adult rehabilitation populations [54]. Generally, parents reported larger improvements in all assessed HRQOL scales than the children themselves. Since children with cancer are at an increased risk for impairments of strength, posture, dexterity, and flexibility due to the different treatment regimens [12], physical therapy is a core element of pediatric oncological rehabilitation. In our sample, physiotherapy was the most frequently applied intervention, with a median number of 36 treatments per patient. Physiotherapeutic treatment consisted of a wide range of individual and group activities, including exercises for strength, dexterity, and mobility, but also hydro gymnastics, wall climbing, and multiple outdoor activities. Additionally, almost all patients took part in occupational therapy, which aimed to improve fine motor skills, sensory motor skills, and visual motor skills. The improvement in the patients’ physical functioning was also underscored by the significant improvement in mean walking distance and general functional status reflected in the Karnofsky score. 

Furthermore, core elements of the rehabilitation treatment were the psychological therapies, which included individual and group sessions, family-oriented interventions, cognitive training, as well as guided relaxation training. Patients received a median of 10.5 sessions during their inpatient treatment. While previous studies have highlighted the effectiveness of psychosocial interventions to improve children’s anxiety, depressive symptoms, and overall quality of life, as well as physical symptoms during active cancer treatment [55], research on psychosocial interventions in pediatric rehabilitation remains scarce. Thus, our study adds to the existing literature, highlighting the beneficial effects of integrated multidisciplinary inpatient treatment programs to improve the psychosocial health of pediatric cancer survivors. 

To evaluate the impact of observed improvements across cancer types, we analyzed how many patients showed clinically important improvements (i.e., above the MCID). The MCID describes the smallest difference in a score that has been perceived as a meaningful improvement by a group of patients [56]. Since MCIDs are only established for the total score of the PedsQL Generic Module, our analysis of MCIDs was limited to this score. While the percentage of patients with clinically meaningful improvements differed across cancer types, these differences were not statistically significant. However, patients with neoplasms of the soft tissue showed the highest percentage of clinically important differences based on both parents’ and children’s ratings. 

The present study has some limitations. For one, parental emotional distress was not assessed in the routine assessment at the rehabilitation center. Since parental distress is discussed as a potential confounder for the discrepancy between parental proxy ratings and children’s self-assessment, the inclusion of standardized measures for anxiety or depression for parents could have helped to explain variance. Secondly, the study is limited by its observational nature and the fact that no control group was available. Since pediatric cancer patients in Austria have the right to receive rehabilitation treatment, which is currently only available as an in-patient treatment, randomization to a control treatment was not possible legally. Nevertheless, we consider the sample to be representative of the target population, since we included a relatively large number of unselected patients. Thirdly, there is no instrument available that has been designed for pediatric cancer survivors specifically. Thus, the included scales may not be perfectly suited to evaluate rehabilitation programs for children who completed cancer therapy. However, the PedsQL is the most frequently used questionnaire in the field [57,58,59], which allows a certain comparability to other studies.

## 5. Conclusions

The results of our analysis show large and clinically relevant improvements in functioning and symptoms among survivors of childhood cancer undergoing multidisciplinary inpatient rehabilitation. This indicates that family-oriented oncological rehabilitation can effectively improve the bio-psycho-social health of pediatric cancer survivors, and may help to reduce long-term effects of cancer and its treatment and thus facilitate re-integration into daily life. The performance score led to substantially better agreement with regard to change assessment between children and parents, and should thus be integrated in future studies. Future research should also focus on the long-term effects of inpatient rehabilitation treatment for cancer survivors. 

## Figures and Tables

**Figure 1 cancers-14-04855-f001:**
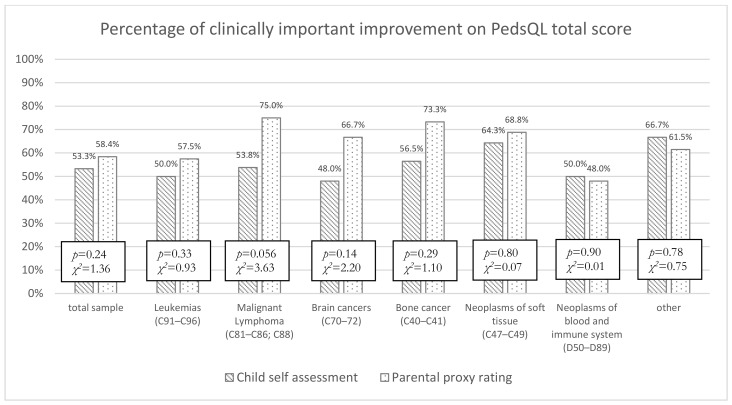
Percentage of patients with clinically relevant improvement based on the PedsQL total score presented for each cancer entity for patient self-report and proxy reports.

**Table 1 cancers-14-04855-t001:** Treatment modalities: overview over therapeutic measures per patient during the rehabilitation.

Treatment Modality	*n*	(%)	Treatment Frequency per Patient	
			Median	IQR
Guidance and treatment by physician	236	(100.0%)	6	5–7
Nursing procedures	236	(100.0%)	4	3–8
Speech therapy	206	(87.3%)	2	1–4
Nutritional counseling	236	(100.0%)	5	4–6
Social counseling	226	(95.8%)	2	1.75–2
Physiotherapy	236	(100.0%)	36	30–42
Massage	179	(75.8%)	3	2–4
Functional occupational therapy	235	(99.6%)	13	10–16
Animal assisted therapy	55	(23.3%)	1	1–2
Psychological therapy	236	(100.0%)	10.5	9–13
Pedagogical interventions	161	(68.2%)	3	2–5

*n*: number of patients who received specific treatment at least once (percentage of patients of the whole collective); IQR: interquartile range.

**Table 2 cancers-14-04855-t002:** Patient characteristics pediatric oncological rehabilitation (*n* = 236).

Mean age (SD)	11.0	(4.3)
(range)	(5–22)	
Sex		
Male	139	58.9%
Female	97	41.1%
Mean body mass index, BMI (SD)	20.0	(5.0)
(range)	(12.7–33.1)	
Mean initial walking range in meter (SD)	519.6	(118.6)
(range)	(0–745)	
Karnofsky Performance Score		
High level of functioning (80–100%)	183	77.5%
Medium level of functioning (50–80%)	49	20.8%
Low level of functioning (0–50%)	2	0.8%
Missing information	2	0.8%
Cancer Entities		
Leukemias (C91–C96)	83	35.2%
Malignant lymphoma (C81–C86; C88)	46	19.5%
Brain cancers (C70–72)	32	13.6%
Bone cancer (C40–C41)	27	11.4%
Neoplasms of soft tissue (C47–C49)	18	7.6%
Neoplasms of blood and immune system (D50–D89)	15	6.4%
Other	15	6.4%

**Table 3 cancers-14-04855-t003:** Self-reported quality of life before and after rehabilitation (child report).

	T1 (*n* = 236)		T2 (*n* = 182)		diff		
	Mean	(SD)	Mean	(SD)	(T2-T1)	*p*-Value	η^2^
PedsQL generic core scale							
Physical functioning	73.9	19.6	81.4	15.6	7.5	<0.001	0.183
Emotional functioning	76.2	20.7	81.5	18.6	5.3	<0.001	0.088
Social functioning	79.9	16.5	84.2	17.4	4.3	0.005	0.043
School functioning	71.1	18.4	76.6	19.0	5.5	<0.001	0.116
Psychosocial functioning	75.8	14.6	80.8	15.1	5.0	<0.001	0.123
Total score	75.1	14.5	81.0	13.8	5.9	<0.001	0.180
PedsQL Cancer Module							
Pain and hurt	76.9	23.3	80.2	23.2	3.3	0.033	0.025
Nausea	79.1	18.1	81.8	16.7	2.7	0.025	0.028
Procedural anxiety	69.4	33.3	76.1	31.5	6.7	0.035	0.025
Treatment anxiety	86.3	20.5	87.5	20.9	1.2	0.27	0.007
Worry	77.7	24.9	79.6	21.6	1.9	0.056	0.020
Cognitive problems	76.4	20.8	79.0	21.3	2.6	0.035	0.024
Perceived physical appearance	79.6	22.4	80.2	22.0	0.6	0.88	<0.01
Communication	78.6	23.7	83.2	22.3	4.6	0.010	0.037
Total score	78.0	14.2	80.9	14.8	2.9	0.004	0.045

SD—standard deviation, *p*-value—level of significance, diff—mean difference; η^2^—effect size (partial eta square); effect size is considered small for η^2^ > 0.01, medium for η^2^ > 0.06, and large for η^2^ > 0.14.

**Table 4 cancers-14-04855-t004:** Self-reported quality of life of the pediatric patients before and after rehabilitation (parent report).

	T1 (*n* = 478)		T2 (*n* = 478)		diff		
	Mean	(SD)	Mean	(SD)	(T2-T1)	*p*-Value	η^2^
PedsQL generic core scale							
Physical functioning	70.7	20.5	78.0	19.4	7.3	<0.001	0.215
Emotional functioning	64.7	18.6	74.3	17.9	9.6	<0.001	0.276
Social functioning	73.3	18.7	78.6	18.9	5.3	<0.001	0.124
School functioning	69.1	19.6	76.6	19.1	7.5	<0.001	0.141
Psychosocial functioning	69.0	14.8	76.5	15.8	7.5	<0.001	0.274
Total score	69.6	15.2	77.0	15.8	7.4	<0.001	0.305
PedsQL Cancer Module							
Pain and hurt	72.0	24.9	78.6	22.6	6.6	<0.001	0.093
Nausea	74.3	19.5	79.0	18.4	4.7	<0.001	0.084
Procedural anxiety	57.1	36.1	68.7	35.0	11.6	<0.001	0.169
Treatment anxiety	79.4	23.5	84.5	22.1	5.1	<0.001	0.070
Worry	76.3	24.9	82.2	23.0	5.9	<0.001	0.090
Cognitive problems	65.5	22.2	68.5	22.8	3.0	0.009	0.020
Perceived physical appearance	77.0	24.7	81.3	22.0	4.3	<0.001	0.048
Communication	57.7	28.8	57.4	31.1	−0.2	0.84	<0.01
Total score	69.7	14.7	74.8	15.5	5.1	<0.001	0.176

SD—standard deviation, *p*-value—level of significance, diff—mean difference; η^2^—effect size (partial eta square); effect size is considered small for η^2^ > 0.01, medium for η^2^ > 0.06, and large for η^2^ > 0.14.

**Table 5 cancers-14-04855-t005:** Associations of the children’s self-report and parental proxy ratings before rehabilitation (t0), stratified by patient–mother and patient–father dyads.

	Patient		Mother		Father		Patient–Mother Dyad			Patient–Father Dyad			Mother–Father Dyad		
	m	SD	m	SD	m	SD	r_ICC_	95% CI	*p*	r_ICC_	95% CI	*p*	r_ICC_	95% CI	*p*
Functioning scales															
PF	78.2	19.1	72.0	18.4	68.1	19.8	0.69	0.51–0.80	<0.001	0.65	0.37–0.79	<0.001	0.76	0.64–0.84	<0.001
EF	76.7	19.7	60.8	19.9	64.6	19.7	0.58	0.08–0.78	<0.001	0.57	0.25–74	<0.001	0.86	0.79–0.91	<0.001
SF	81.7	14.2	75.1	15.8	74.5	15.1	0.53	0.28–0.69	<0.001	0.59	0.34–0.74	<0.001	0.78	0.66–0.85	<0.001
SchF	74.9	16.5	69.3	19.2	69.8	16.9	0.65	0.47–0.77	<0.001	0.59	0.38–0.73	<0.001	0.80	0.70–0.87	<0.001
PsF	77.8	13.4	68.4	14.7	69.6	13.4	0.64	0.24–0.81	<0.001	0.63	0.29–0.79	<0.001	0.86	0.79–0.91	<0.001
TotF	77.9	13.1	69.6	14.3	69.1	13.9	0.66	0.31–0.81	<0.001	0.63	0.25–0.80	<0.001	0.84	0.77–0.90	<0.001
Symptom scales															
Pain	80.4	21.8	69.3	24.0	71.6	25.7	0.73	0.48–0.85	<0.001	0.66	0.46–0.78	<0.001	0.81	0.71–0.87	<0.001
Nausea	80.1	17.5	70.0	20.1	69.8	20.3	0.61	0.31–0.77	<0.001	0.46	0.14–66	<0.001	0.82	0.71–0.88	<0.001
Prod Anx	67.7	33.2	59.5	35.0	63.2	33.0	0.84	0.74–0.90	<0.001	0.78	0.66–0.85	<0.001	0.82	0.74–0.88	<0.001
Treat Anx	87.2	19.6	78.1	23.1	80.5	21.3	0.61	0.38–0.75	<0.001	0.52	0.29–0.68	<0.001	0.77	0.66–0.85	<0.001
Worry	80.4	24.6	71.5	25.7	72.5	23.2	0.67	0.48–0.78	<0.001	0.64	0.45–0.76	<0.001	0.79	0.69–0.86	<0.001
Cogn	79.3	19.9	65.2	22.4	63.3	20.1	0.49	0.15–0.68	<0.001	0.40	0.03–0.63	<0.001	0.61	0.41–0.74	<0.001
Perc Attr	85.0	18.8	70.2	25.2	72.8	22.1	0.50	0.16–0.69	<0.001	0.57	0.26–0.74	<0.001	0.77	0.65–0.85	<0.001
Comm	77.8	24.5	53.7	28.7	52.8	29.4	−0.36	−0.97–0.08	0.98	–0.17	−0.59–0.16	0.86	0.65	0.48–0.77	<0.001
TotSym	79.5	13.4	69.6	14.3	69.1	13.9	0.52	0.15–0.71	<0.001	0.47	0.09–0.68	<0.001	0.84	0.77–0.90	<0.001

m—mean score; SD—standard deviation; r_ICC_—intraclass-correlation coefficient; CI—confidence interval; PF—physical functioning; EF—emotional functioning; SF—social functioning; SchF—school functioning; PsF—psychosocial functioning; TotF—total functioning score; Prod Anx—procedural anxiety; Treat Anx—treatment anxiety; Cogn—cognitive problems; Perc Attr—perceived physical appearances; comm—communication; TotSym—total symptom score; r_ICC_ < 0.5 = poor, 0.5–0.75 = moderate, 0.75–0.9 = good, and >0.9 = excellent reliability (Koo and Li, 2016) [38].

**Table 6 cancers-14-04855-t006:** Associations of the children’s self-report and parental proxy ratings after rehabilitation (T1), stratified by patient–mother and patient–father dyads.

	Patient		Mother		Father		Patient–Mother Dyad			Patient–Father Dyad			Mother–Father Dyad		
	m	SD	m	SD	m	SD	r_ICC_	95% CI	*p*	r_ICC_	95% CI	*p*	r_ICC_	95% CI	*p*
Functioning scales															
PF	83.5	14.5	80.4	16.3	78.3	18.4	0.86	0.76–0.91	<0.001	0.78	0.58–0.88	<0.001	0.86	0.78–0.91	<0.001
EF	79.9	20.0	74.5	20.3	73.9	20.5	0.77	0.62–0.86	<0.001	0.76	0.60–0.86	<0.001	0.88	0.81–0.93	<0.001
SF	85.3	16.4	82.9	16.9	80.5	17.6	0.84	0.73–0.90	<0.001	0.80	0.62–0.89	<0.001	0.82	0.71–0.89	<0.001
SchF	78.3	19.5	78.4	20.5	77.1	19.8	0.85	0.75–0.91	<0.001	0.81	0.68–0.88	<0.001	0.79	0.67–0.87	<0.001
PsF	81.0	15.1	78.6	17.1	77.3	17.1	0.87	0.78–0.92	<0.001	0.83	0.70–0.90	<0.001	0.86	0.77–0.91	<0.001
TotF	81.9	13.9	79.2	15.8	77.7	16.2	0.88	0.80–0.93	<0.001	0.85	0.69–0.92	<0.001	0.88	0.81–0.93	<0.001
Symptom scales															
Pain	80.8	24.5	75.4	25.1	74.8	24.0	0.78	0.64–0.87	<0.001	0.87	0.77–0.92	<0.001	0.89	0.83–0.94	<0.001
Nausea	82.5	17.9	78.1	22.3	77.1	22.1	0.65	0.40–0.80	<0.001	0.74	0.55–0.86	<0.001	0.84	0.72–0.90	<0.001
Prod Anx	75.4	33.3	71.0	35.5	74.2	32.2	0.88	0.81–0.93	<0.001	0.91	0.85–0.95	<0.001	0.86	0.77–0.91	<0.001
Treat Anx	86.9	23.1	81.1	25.9	83.6	24.8	0.75	0.59–0.84	<0.001	0.82	0.71–0.89	<0.001	0.85	0.76–0.91	<0.001
Worry	81.0	21.8	79.0	23.9	74.1	26.0	0.81	0.68–0.88	<0.001	0.77	0.62–0.87	<0.001	0.79	0.68–0.87	<0.001
Cogn	80.7	20.5	68.4	23.9	65.1	22.3	0.64	0.32–0.80	<0.001	0.40	0.01–0.64	0.004	0.50	0.21–0.69	<0.001
Perc Attr	84.0	20.2	80.2	23.8	75.8	25.4	0.74	0.58–0.84	<0.001	0.64	0.41–0.79	<0.001	0.82	0.72–0.89	<0.001
Comm	81.2	25.3	54.1	32.3	46.6	30.2	−0.36	−10.1–0.13	0.96	0.01	−0.27–0.27	0.47	0.46	0.15–0.66	0.005
TotSym	81.4	16.0	79.2	15.8	78.8	15.1	0.79	0.65–0.87	<0.001	0.82	0.69–0.89	<0.001	0.88	0.81–0.93	<0.001

m—mean score; SD—standard deviation; r_ICC_—intraclass-correlation coefficient; CI—confidence interval; PF—physical functioning; EF—emotional functioning; SF—social functioning; SchF—school functioning; PsF—psychosocial functioning; TotF—total functioning score; Prod Anx—procedural anxiety; Treat Anx—treatment anxiety; Cogn—cognitive problems; Perc Attr—perceived physical appearances; comm—communication; TotSym—total symptom score; r_ICC_ < 0.5 = poor, 0.5–0.75 = moderate, 0.75–0.9 = good, and >0.9 = excellent reliability (Koo and Li, 2016) [38].

**Table 7 cancers-14-04855-t007:** Comparison of agreement in mean improvement during therapy between children and parents using the delta and the performance score.

	Delta (*n* = 149)		Performance Score (*n* = 149)	
	r	*p*-Value	r	*p*-Value
PedsQL generic core scale				
Physical functioning	0.51	<0.001	0.59	<0.001
Emotional functioning	0.36	<0.001	0.52	<0.001
Social functioning	0.37	<0.001	0.51	<0.001
School functioning	0.45	<0.001	0.60	<0.001
Psychosocial functioning	0.53	<0.001	0.65	<0.001
Total score	0.59	<0.001	0.68	<0.001
PedsQL Cancer Module				
Pain and hurt	0.28	0.004	0.42	<0.001
Nausea	0.19	0.10	0.40	<0.001
Procedural anxiety	0.20	0.043	0.60	<0.001
Treatment anxiety	0.24	0.016	0.50	<0.001
Worry	0.09	0.37	0.33	0.001
Cognitive problems	0.26	0.009	0.29	0.003
Perceived physical appearance	0.43	<0.001	0.49	<0.001
Communication	<0.01	0.99	−0.09	0.35
Total score	0.45	<0.001	0.60	<0.001
Mean r-scores	0.34		0.50	

r—Pearson correlation coefficient; delta—mean difference during rehabilitation (T2-T1); performance score—adjusted delta (T2 + (T2-T1)); PF—physical functioning; EF—emotional functioning; SF—social functioning; SchF—school functioning; PsF—psychosocial functioning; TotF—total functioning score; Prod Anx—procedural anxiety; Treat Anx—treatment anxiety; Cogn—cognitive problems; Perc Attr—perceived physical appearances; Comm—communication; TotSym—total symptom score; r < 0.3 = negligible, r = 0.3–0.5 low, 0.5–0.7 moderate, 0.7–0.9 high, and >0.9 excellent correlation (Hinkle, Wiersma, and Jurs, 2003) [40].

## Data Availability

Data are part of the patients’ medical records at VAMED Rehabilitation Center Leuwaldhof (Austria). They were anonymized for statistical analysis. The datasets used and/or analyzed during the current study are available from the corresponding author upon reasonable request. Computer-based health evaluation software (CHES) was used for collection of electronic patient-reported outcome data. This software application is described under: https://ches.pro/. (accessed on 10 March 2022).

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
