# Peer review of "Large Improvements in Health-Related Quality of Life and Physical Fitness during Multidisciplinary Inpatient Rehabilitation for Pediatric Cancer Survivors"

_cancers, 2022, doi:10.3390/cancers14194855_

Round 1

Reviewer 1 Report

David Riedl and collaborators present an observational, longitudinal, mono-centric study in long-term pediatric (and young adults) patients survivors of a form of pediatric cancer who undergo a rehabilitation program with their parents/proxies. The study has several strengths: 1) Large number of patients representative of the spectrum of diseases in this age-period; 2) The hypothesis is relevant and the methodology to respond the questions raised by the authors is rigorous; 3) The manuscript is very well written, the description of the methodology is clear, the results are of interest for the scientific community and the discussion is rich and addresses several current points of discrepancy among researches; lastly, they recognize the limitations of the study, identify areas of improvement and suggestions to address future research questions, based on the large experience in this field from the authors.

I have only one major comment and some minor comments.

Major comment:

I have not seen in the manuscript whether the patients and proxies were asked for an informed consent to participate. This is a prospective study, where personal and health information is collected to asses the effect of an intervention and the results are intended for publication purposes. Moreover, the authors mention that the datasets are available upon request; I understand that they may want to facilitate this information for other researchers outside the leading group (even if anonymized) if they are asked for. The tests to evaluate the intervention are not part of the routine practice, to my understanding. Therefore, an informed consent is necessary, not only the approval of the Institution (which is described at the bottom of the manuscript). The authors need to confirm if this was not done, and if not, explain the reasons for not having done so (e.g. country specific legislation).

Minor comments:

- In general the manuscript is very long; some sections are overdetailed and there may be parts that can be omitted to focus more on the research question and objectives of the study. I give here some examples:

* Introduction: Page 2 lines 48 to 57. This is a general description of pediatric oncology, incidence, distribution of diseases, etc. The reader interested in this manuscript is aware of this information and likely not necessary to introduce the matter of the research. Similar, paragraph from lines 58-65 is very detailed and well know. There is no need to describe what physical impairment or decreases functioning is with so many examples. I.e: physical impairment (such as fatigue, digestive symptoms, cardi- 61 otoxicity, neurocognitive and hormonal deficits, impaired sexual development, and infer- 62 tility) and decreased functioning (including neuropsychological functioning; impact on 63 autonomy, school, and social life). Sentences like “However, the use of proxy reports is not entirely unproblematic” are difficult to understand, please use simpler terms. To combine to negative terms in the same sentence and so close make understanding difficult (not entirely unproblematic). The 3 other paragraphs from Line 66 to 105 are also overdetailed; while the information presented there is relevant, it is too much and it is difficult for the reader to know exactly what the point or research and problem is. Please shorten as much as possible this section, shorter sentences and clear message.

- Page 6, Line 212. The authors mention the following: Patients with missing data at T2 reported significantly worse school functioning 212 (69.6 vs. 77.0 points; F(1, 232) = 6.75, p = 0.010, η2 = 0.029) and procedural anxiety (59.5 vs. 213 72.3 points; F(1, 232) = 6.04, p = 0.015, η2 = 0.026). Because the number of patients with missing data is not that high (236 – 182= 54) and they compare it with the others who complete T2, I find risky to make that kind of comparisons and try to get conclusions from it. Someone can get the impression that patients with poorer school functioning and more anxiety are less prone to fill T2, but the question is what this really means. Please explain or remove.

- Section 3.4. It is not described the number of dyads at T2. Please add.

- Page 11, Line 322. The authors mention “A comparison of the children’s and parents’ performance scores yielded substantially improved agreement for all assessed subscales, with at least moderate correlations (r = 323 0.29-0.68)”. I understand that the authors want to correct the ceiling effect due to baseline differences as it is explained in the methods by using the performance score T2 and this is very fine strategy. The way this is written, may lead to the interpretation from the reader that by using this technique, one can get more agreement when comparing different tests with missing data, and therefore ‘over-estimating’ the positive effect of the association; while in reality, what the technique permits is to really find whether differences exist between the compared items reducing the bias (both positive and negative differences). It should be stressed in the manuscript/discussion that this technique may help to identify real differences (positive and negative or neutral) by avoiding the risk of using delta in this particular situations so not everybody is tempted to use this technique to find more ‘positive’ results.

- Not sure why the authors underline certain words (rehabilitation goals, 6 minute walk). Unless really necessary, please keep the same outlook for all the text.

- Page 4 Line 135. They mention six functioning subscales (physical, emotional, social, school, psychosocial), but in brackets there are only 5. Please add the missing one.

- Page 5 Line 204. The percentage of brain tumors is not the same between the text and table 1 (11.4 Vs 13.6). Please correct.

Author Response

Reviewer 1

David Riedl and collaborators present an observational, longitudinal, mono-centric study in long-term pediatric (and young adults) patients survivors of a form of pediatric cancer who undergo a rehabilitation program with their parents/proxies. The study has several strengths: 1) Large number of patients representative of the spectrum of diseases in this age-period; 2) The hypothesis is relevant and the methodology to respond the questions raised by the authors is rigorous; 3) The manuscript is very well written, the description of the methodology is clear, the results are of interest for the scientific community and the discussion is rich and addresses several current points of discrepancy among researches; lastly, they recognize the limitations of the study, identify areas of improvement and suggestions to address future research questions, based on the large experience in this field from the authors.

Thank you for this positive feedback and your careful reading of our manuscript. We have revised the manuscript based on your comments which helped us to further increase the readability of the paper.

Major comment:

I have not seen in the manuscript whether the patients and proxies were asked for an informed consent to participate. This is a prospective study, where personal and health information is collected to asses the effect of an intervention and the results are intended for publication purposes. Moreover, the authors mention that the datasets are available upon request; I understand that they may want to facilitate this information for other researchers outside the leading group (even if anonymized) if they are asked for. The tests to evaluate the intervention are not part of the routine practice, to my understanding. Therefore, an informed consent is necessary, not only the approval of the Institution (which is described at the bottom of the manuscript). The authors need to confirm if this was not done, and if not, explain the reasons for not having done so (e.g. country specific legislation).

We apologize for this shortcoming; we completely agree with reviewer that for this kind of study an informed consent is necessary. At the rehabilitation center where the data was collected, patients and / or their legal guardians signed an informed consent before first data collection. All collected data (both PROMs and CROMs) that were analyzed in the study were collected as part of the clinical routine and no additional data was collected solely for the study. The ICF and study protocol was approved by the approved by the Ethics Commission of the State of Salzburg (Austria).

We added this information in the manuscript on page 3, line 106-107 and in the according informed consent statement form at the end of the manuscript.

Minor comments:

- In general the manuscript is very long; some sections are overdetailed and there may be parts that can be omitted to focus more on the research question and objectives of the study. I give here some examples:

Thank you for the suggestions. We have revised the manuscript based on your suggestions and the introduction was shortened by eleven lines. More detailed information is given below. To allow readability, all changes to the manuscript were highlighted in yellow.

* Introduction: Page 2 lines 48 to 57. This is a general description of pediatric oncology, incidence, distribution of diseases, etc. The reader interested in this manuscript is aware of this information and likely not necessary to introduce the matter of the research.

Similar, paragraph from lines 58-65 is very detailed and well know. There is no need to describe what physical impairment or decreases functioning is with so many examples. I.e: physical impairment (such as fatigue, digestive symptoms, cardi- 61 otoxicity, neurocognitive and hormonal deficits, impaired sexual development, and infer- 62 tility) and decreased functioning (including neuropsychological functioning; impact on 63 autonomy, school, and social life).

While we agree with the reviewer that this general information is well known to experts in pediatric oncology, we felt the need to give at least some background for interested readers who may be unfamiliar with the field (especially since Cancers mainly publishes research about adult oncology).

However, we understand that these passages may be a bit lengthy for readers with a pediatric background and have thus shortened the description of physical and psychosocial impairments on page 2, lines 59-62.

Sentences like “However, the use of proxy reports is not entirely unproblematic” are difficult to understand, please use simpler terms. To combine to negative terms in the same sentence and so close make understanding difficult (not entirely unproblematic).

The sentence was changed to “However, the use of proxy reports is somehow problematic” to increase readability.

The 3 other paragraphs from Line 66 to 105 are also overdetailed; while the information presented there is relevant, it is too much and it is difficult for the reader to know exactly what the point or research and problem is. Please shorten as much as possible this section, shorter sentences and clear message.

The three paragraphs from line 66 to 105 were revised and shortened as suggested.

- Page 6, Line 212. The authors mention the following: Patients with missing data at T2 reported significantly worse school functioning 212 (69.6 vs. 77.0 points; F(1, 232) = 6.75, p = 0.010, η2 = 0.029) and procedural anxiety (59.5 vs. 213 72.3 points; F(1, 232) = 6.04, p = 0.015, η2 = 0.026). Because the number of patients with missing data is not that high (236 – 182= 54) and they compare it with the others who complete T2, I find risky to make that kind of comparisons and try to get conclusions from it. Someone can get the impression that patients with poorer school functioning and more anxiety are less prone to fill T2, but the question is what this really means. Please explain or remove.

The paragraph was removed as suggested (now page 5, line 200).

- Section 3.4. It is not described the number of dyads at T2. Please add.

Data of available dyads for T2 was added (page 8, line 263-266

- Page 11, Line 322. The authors mention “A comparison of the children’s and parents’ performance scores yielded substantially improved agreement for all assessed subscales, with at least moderate correlations (r = 323 0.29-0.68)”. I understand that the authors want to correct the ceiling effect due to baseline differences as it is explained in the methods by using the performance score T2 and this is very fine strategy. The way this is written, may lead to the interpretation from the reader that by using this technique, one can get more agreement when comparing different tests with missing data, and therefore ‘over-estimating’ the positive effect of the association; while in reality, what the technique permits is to really find whether differences exist between the compared items reducing the bias (both positive and negative differences). It should be stressed in the manuscript/discussion that this technique may help to identify real differences (positive and negative or neutral) by avoiding the risk of using delta in this particular situations so not everybody is tempted to use this technique to find more ‘positive’ results.

Thank you for this valuable suggestion; we have added this information in the discussion section (page 13, line 400-404)

- Not sure why the authors underline certain words (rehabilitation goals, 6 minute walk). Unless really necessary, please keep the same outlook for all the text.

Thank you for highlighting this. The idea was to highlight the assessed PROMs and CROMs. To increase readability underlining was removed.

- Page 4 Line 135. They mention six functioning subscales (physical, emotional, social, school, psychosocial), but in brackets there are only 5. Please add the missing one.

This was a typo, the PedsQL actually consists of five subscales. The number was corrected in the manuscript.

- Page 5 Line 204. The percentage of brain tumors is not the same between the text and table 1 (11.4 Vs 13.6). Please correct.

Thank you for highlighting this typo; the number in the text was corrected.

Reviewer 2 Report

The manuscript is interesting and well written. However some clarifications are necessary. To my knowledge, more precise information and discussions would be needed regarding the starting sample and above all its characteristics. Do any of the participants have physical and cognitive limitations such as, for example, cognitive retardation, amputations or outcomes of surgical or drug interventions?what treatments have they benefited from during active therapy? have they already benefited from these?it is not clear to me if at the baseline there are any differences on the basis of the pathology? I find it very strange that the initial quality of life is the same between a recovered from leukemia rather than from a brain tumor. however, even the differences not found should be discussed and commented on.

Author Response

Comments:The manuscript is interesting and well written.

Reply: Thank you for this encouraging comment.

Comments: 

However some clarifications are necessary. To my knowledge, more precise information and discussions would be needed regarding the starting sample and above all its characteristics. Do any of the participants have physical and cognitive limitations such as, for example, cognitive retardation, amputations or outcomes of surgical or drug interventions?

Reply: Thank you for giving us a possibility to clarify on this matter. The reviewer is absolutely right to point out that children who survived cancer diagnosis and treatment and come to the rehabilitation center often have to bear substantial physical and cognitive limitations. In the rehabilitation center they do, for example, treat children and AYAs after brain tumors, who may suffer from impaired vision or even are blind, or who suffer from ataxia or hemiplegia, while others children with the same diagnosis did not end up with such severe long-term consequences of the treatment. Since the therapy outcomes and adverse events are very diverse and do not only differ from cancer to cancer, but also strongly differ within cancer entity, we used the Karnofsky Performance Score to objectify the degree of impairment across groups. Furthermore, the specific impairment was assessed with the corresponding PedsQL scales and / or observer-ratings of physical fitness.

Comments: what treatments have they benefited from during active therapy? have they already benefited from these?

Reply: We have no complete information on the treatment children had received during active cancer treatment. However, in Austria, treatment usually includes multidisciplinary in- or outpatient treatment with by physicians, nurses, psychologist and physical therapists, which is fully covered by social security providers. Since treatment may vary between hospitals and children’s age, in this study we have focused on the self-, proxy- and physician-rated health at the start of rehabilitation as means to quantify the children’s health state.

Comments: it is not clear to me if at the baseline there are any differences on the basis of the pathology? I find it very strange that the initial quality of life is the same between a recovered from leukemia rather than from a brain tumor. however, even the differences not found should be discussed and commented on.

Reply: 

Thank you very much for this helpful observation. We have revised the analyses for baseline comparison between pathologies. While we initially only included a comparison between the children’s PedsQL scores (where no significant differences were observed), we also included analyses of the parents PedsQL scores and the Karnofsky Performance Score. In these analyses we found that specifically children with sarcomas had lower Performance scores than children with leukemia or malignant lymphomas and that parents of children with sarcomas, leukemia, brain tumors, and malignant reported worse pain scores than parents of children with neoplasms of blood and immune system and with other malignancies. We added the results on page 6, line 220 – 230.

Additionally, we have revised the discussion section and added additional literature in comparison to our findings on page 12, line 364-369.

We thank the reviewer for pointing this issue out and thus allowing us to further increase the quality of our manuscript. 

Round 2

Reviewer 1 Report

I thank the authors for addressing my comments. I have no further comments. 

Reviewer 2 Report

Many thanks for your revisions